# The Monetary Benefits of Reducing Emissions of Dioxin-like Compounds—Century Poisons—Over Half a Century: Evaluation of the Benefit per Ton Method

**DOI:** 10.3390/ijerph19116701

**Published:** 2022-05-30

**Authors:** Je-Liang Liou, Han-Hui Chen, Pei-Ing Wu

**Affiliations:** 1The Center for Green Economy, Chung-Hua Institution for Economic Research, Taipei City 10672, Taiwan; jlliou@cier.edu.tw; 2Department of Land Economics, National Chengchi University, Taipei City 11605, Taiwan; gba0852@gmail.com; 3Department of Agricultural Economics, National Taiwan University, Taipei City 10617, Taiwan

**Keywords:** health benefit, impact pathway, lifetime average daily exposure, value transfer method, multiple media transport, value of a statistical life

## Abstract

The objective of this study is to evaluate the monetary value of health benefits following reductions in century poison dioxin-like compounds for people aged 0–14 years old, 15–64 years old, and persons 65 years or over in Taiwan. The benefit per ton (BPT) method is employed to estimate the monetary value of the benefits of such a reduction from 2021 to 2070 for different age groups in different regions. The results indicate a BPT of US$837,915 per gram of dioxin each year. The results further show that for Taiwan as a whole, the net BPT per gram of dioxin reduction from 2021 to 2025 is US$704 for children, US$42,761 for working-age adults, US$34,817 for older adults, and US$78,282 overall. Reductions in dioxin-like compounds from 2051–2070 will generate 83.93% of the net BPT for the entire country. This is approximately five times the net BPT of emissions reduction from 2021 to 2025. The monetary benefits evaluated in this study indicate that the prevention of health losses caused by the spread and diffusion of dioxin-like compounds have increased significantly. This implies that action must be taken now, along with continued vigilance, to address emission reductions.

## 1. Introduction

Polychlorinated dibenzo-p-dioxins (PCDDs), polychlorinated dibenzofurans (PCDFs), and related dioxin-like polychlorinated biphenyl (PCB) compounds are typical persistent organic pollutants (POPs). They are mainly generated by industrial processes, the combustion of hydrocarbons, the incineration of solid residues or wastes, combustion in a cement reactor, etc. [1]. The hazard posed by dioxins has been continually assessed in the past and continues to be studied by the health or environmental departments of many individual countries, such as the United States Environmental Protection Agency (US EPA), Health and Welfare Canada, and the Health Council of the Netherlands, and by international agencies, such as the World Health Organization (WHO), the European Standard (for the measurement of PCDD and PCDF), and the European Food Safety Authority [2,3,4].

Taiwan is no exception. In Taiwan, dioxin-like compounds are known as “century poisons”, and thus, the Environmental Protection Administration, Executive Yuan, R.O.C., Taiwan (Taiwan EPA), legislated the “Waste Incinerator Dioxin Control and Emission Standards” in 1997 [5]. However, the year after this standard came into effect, the first serious incident with a solid-waste incinerator occurred in Taipei, the capital of Taiwan. Since then, poison incidences have been continually observed in ducks, duck eggs, sheep, and different kinds of food. Food safety has become a source of panic for people in Taiwan due to the contamination of products with dioxin-like compounds. The Taiwan EPA, similar to all other agencies or institutes around the world, has consistently devoted effort to eliminating dioxin-like compound emissions from all sources, mitigating their negative effects on the human body and reducing the level of background dioxin contamination in the environment.

To appropriately implement an effective policy, the assessment of changes in the levels of dioxin-like POPs under various activities or circumstances and an examination of the impact of dioxin emissions at the country level must be conducted. New technologies may emerge from these assessments. Examples of this can be found from the study Karademir [6] conducted to evaluate a hazardous and medical waste incinerator in Turkey and from the study by Nguyen et al. [7] which assessed the formation of PCDDs/PCDFs in open dumping sites in various countries in Asia. As dioxin-like compounds naturally combine with other POPs during industrial activities, attention has also been given to industrial activity, a major generator of dioxin-like compounds, to develop an appropriate indicator for the monitoring and control of dioxin-like POPs [8]. Various technologies have been proposed to reduce dioxin emissions, and life-cycle assessments have also been conducted [9].

Furthermore, regarding the physical dimensions of emissions of dioxin-like compounds, past studies have emphasized the measurement of the different ways in which dioxin-like compounds diffuse and potentially spread: inhalation, dermal absorption of soil and dust, and ingestion of soil and dust [10]. For instance, some studies have investigated certain areas or regions for potential PCDD/PCDF contamination through specific pathways. Sediment runoff is one of the channels by which these toxic elements are spread. Thus, the management of sediment is essential for controlling the spread of dioxins [11]. Sundqvist et al. [12] also explored the variation in PCDD/PCDF levels in Baltic surface sediment, as the Baltic Sea is heavily impacted by the atmospheric deposition and freshwater inputs of these compounds.

Dioxins or dioxin-like compounds commonly settle in the air, in water, or on land. The human body is easily affected by each of these pathways. Airborne dioxins can be directly or indirectly inhaled or consumed by humans. Milk from dioxin-contaminated dairy farms is a typical source of ingested dioxins; adults or infants could consume affected dairy products [13,14,15]. Likewise, fish products contaminated with dioxin inevitably have adverse effects on human health if the water is polluted. Fish provide good, high-quality protein for the human body and are a crucial food source for 3 billion people globally. Of those countries dependent on fish, Taiwan has frequently been on the list of the top 25 global producers of marine-caught fish [16]. Concern over contaminated fish has induced scholars to examine the toxicity of fish contaminated by different types of heavy metals and the global variation in the health risks of consuming different quantities of such fish [17,18,19,20,21,22].

Dioxins normally enter the human body through a variety of pathways, including water, soil, and air. Rovira et al. [23] assessed the bodily concentrations of dioxin-related compounds present in the environment along with those of other heavy metals that are transmitted either through the air or through soil among people living in Spain. They found that concentrations were significantly higher for people living within 1 km of a municipal solid waste (MSW) incinerator than those living more than 1 km away. To determine the toxicity of dioxin emissions, a proper indicator must first and foremost be identified, though that is not the only criterion for managing dioxin emissions. Indeed, it is essential to know the impact of exposure to different levels or quantities of dioxin emissions. The impact of dioxins on human health is a major concern for everyone.

The carcinogenic health risk is often used as an indicator to reflect the potential impacts on people who live in areas with different distributions of 95 PCDD/PCDF emissions or concentrations [1,24,25,26,27]. Additionally, a risk assessment has been conducted for dioxin-like non-genotoxic chemical carcinogens, and that assessment found that exposure to such carcinogens may result in non-genotoxic activities and oxidative deoxyribonucleic acid (DNA) damage and may initiate carcinogenesis [28]. Earlier, Melnick et al. [29] and Riahi et al. [30] systematically reviewed the potential activities and suggested the critical factors for the carcinogenic effect of certain non-genotoxic carcinogens.

Later, Domingo et al. [31] specifically evaluated the emission reduction-induced change in the health risks of exposure to PCDD/PCDF emissions for adults and children living within 500 m and 1000 m of a MSW incinerator by comparing the reduction in air emissions of PCDDs/PCDFs from that MSW incinerator before 1998 and after 2000 in Montcada, Spain. The results showed that the health risk decreased for adults by approximately 75% due to the air emissions reduction when the city transitioned from the traditional incinerator used in 1998 to modern incinerator technologies in 2000. A similar decrease in health risk of approximately 68% was found for children. These results indicate that improving incinerator technologies alone can reduce the impact of dioxin-like compounds on health risks to a certain extent, even though background contamination was not taken into account [32]. The results also reveal that environmental chemical contaminants have different impacts on persons of different ages [33,34].

Most previous studies investigated the effects of the physical dimension of dioxin-like compounds. Certain studies have explored the changes in health risks arising from changes in incinerator technology, a major source of dioxin emissions, or structural changes in the manufacturing industry. These outcomes are normally reported either in terms of a specific type of PCDD/PCDF or as a measurement of the concentration of dioxin-like compounds. However, if the physical measurement of dioxin-like compounds is not monetized, no information allowing comparison with the costs of new technologies or structural changes will be available. Nadal et al. [10], Fuster et al. [24], and Shih et al. [35] are among the few who have calculated monetary values in terms of either the value of a statistical life (VSL) or health costs for health assessments. However, their studies were specific to cement plants using sewage sludge as a fuel source in Spain and the reduction in waste dumping at specific sites in Nairobi, Kenya.

The existing literature neither systematically evaluates an entire country nor provides an estimate of the monetary value of the benefits accrued as a counterpart to the monetary cost of reducing emissions of dioxin-like compounds. A systematic review of this kind could be valuable as a reference when such emission reduction campaigns are launched. The health risk from exposure to dioxin-like compounds differs across areas or regions within a country, as many varieties of plants are located in various parts of any given country. Thus, it is essential for decision makers to understand the monetary value of health benefits to establish specific management targets for the reduction in all kinds of PCDD/PCDF emissions in the environment. Monetary values can be used as a benchmark against which costs can be compared when reduction targets for emissions of dioxin-like compounds are proposed. To use VSL to calculate the monetary value of a reduction in the emission of dioxin-like compounds, it is necessary to account for the gross domestic product (GDP) and age composition of the population, as these factors can reflect the population density and productivity of the regions or counties of interest within a country [36,37,38].

Specifically, the purpose of this study is first to evaluate the monetary value of the health benefits of a reduction in emissions of dioxin-like compounds in different regions for people of different ages. The benefit per ton (BPT) method is used to simulate this monetary benefit for people living in the different regions of Taiwan in 2020 who are aged 0–14 (not in the labor force), aged 15–64 years (the majority of the labor force), and aged 65 or above (retired). Second, the BPT method is also used to simulate the monetary value of the benefits of a reduction in emissions of dioxin-like compounds from 2021 to 2050 for the different age groups of the projected population. 2050 is the target year for the reduction of greenhouse gases (GHGs) to a certain level. Reductions in GHG emissions are expected, and dioxin-like compound emissions are expected to be reduced to half of their current level by 2050. Currently, population projections for Taiwan are available until 2070. The monetary value of the benefit of a reduction in emissions of dioxin-like compounds is estimated for 20 years beyond 2050 under the assumption that the dioxin-like compounds are reduced at an ambitious pace between 2051 and 2070.

The rest of the paper is arranged as follows. Section 2 presents the methods of the impact pathway approach for monetizing the benefits of a dioxin emission reduction. Section 3 presents the computation and simulations for all phenomena in the empirical analysis. Section 4 presents the results and discussion. Section 5 provides the conclusions of this study.

## 2. Materials and Methods

### 2.1. Monetary BPT According to the Impact Path Approach

Various methods for monetizing the impact of air pollution have been developed. Among these, one state-of-the-art method is the impact path approach (IPA) [39,40]. From a technological viewpoint, the IPA is a combination of three different evaluation frameworks [41]. The IPA framework is illustrated in Figure 1. When monetary outcomes are the focus of evaluations of the impact of a reduction in dioxin-like compound emissions under the IPA framework, the current spread, diffusion of emissions, evaluation of exposure to carcinogenic compounds, and intake, defined as the lifetime average daily dose (LADD), must be identified first.

The second step is to manage the marginal change in monetary outcomes due to a change in air pollution concentration. That is, the emission of dioxin-like compounds from diffusion and spread of dioxin to LADD for each person is computed. The results from the simulation of diffusion and spread on the left-hand side of Figure 1 should be completed before the health impact assessment is conducted. The third step is to learn about changes in events, e.g., health incidents, that are affected by the concentration of air pollution. Finally, an appropriate method is employed to monetize the health impact of changes in health incidents. The Environmental Benefit Mapping and Analysis Program (BenMAP) developed by the US EPA in 2003 and the Air Benefit and Cost and Attainment Assessment System (ABaCAS) are used in this study. 

BenMAP, developed by the US EPA in 2003, is one of the most popular tools for evaluating the public health impacts of air quality. It contains a set of data for evaluating concentration−response relationships and includes demographic characteristics. In 2012, the US EPA made this PC-based software broadly accessible to the global research community. It then released BenMAP-CE in 2015 [42]. Similar to BenMAP, BenMAP-CE can be used to analyze data at different spatial resolutions with varying levels of complexity on the local, regional, national, and global scales. Both programs have been applied to data for various countries outside the US. Such studies include Chae and Park [43] and Heo et al. [44] for South Korea, Nawahda [45] for Japan, Boldo et al. [46] for Spain, and Manojkumar and Srimuruganandam [47] for India.

One feature of the IPA is that it can synthesize the various impacts of air pollution occurring through different paths in monetary terms. Such results can also enable comparisons of different impacts through the implementation of different control mechanisms. This result can further be integrated into a cost benefit analysis (CBA) framework to assist with rational decision making. However, a disadvantage of employing the IPA is that simulating each of the three steps described above is time consuming, and thus, IPA implementation is costly. To resolve such technical problems, some simpler reduced-form tools, as opposed to tools that implement the full IPA program, have been proposed and applied [40,48,49,50]. Reduced-form tools operate under the assumption that the influence of emission diffusion is uniform. That is, emissions are evenly distributed across all the impacted subjects.

Fann et al. [51] and Fann et al. [52] applied the reduced form of BenMAP developed by the US EPA to measure the BPT from an improvement in air pollution at the national level in the US. Similarly, the BPT from a reduction in dioxin-like compound emissions can be measured with BenMAP. First, the emission concentration of each dioxin in each region is calculated. Then, the results are used to compute the monetary value of the health benefits from each ton of emissions reduced through a health impact function and a conversion of the health impact into monetary terms. Each BPT simulation covers a specific time and area. Thus, each estimated BPT is normally used as a reference for several years until updated information is available. Regarding the policy applications of BenMAP evaluations, the US EPA announced in 2013 that it had produced a national-level technical BPT report for all kinds of air pollutant emissions and used that report in a regulatory impact analysis (RIA). The updated open-source version of the BPTs was made available in 2018 [50].

### 2.2. The Steps of the Simulations Used to Compute the Monetary BPT from Dioxin Emission Reductions

This study adopts the BPT method to simulate the unit benefit from a reduction in emissions of dioxin-like compounds. First, the current spread and diffusion of emissions and the LADD need to be identified. The corresponding expected population risk of death from cancer is then inferred from the health risk coefficients, including those for carcinogenic risk and cancer mortality. VSL calculations are employed to estimate the monetary value of the expected mortality risk. The final result is the health cost per unit of dioxin-like compound emissions or the benefit per unit of dioxin-like compound emissions reduced under an assumption of uniformity.

Under the framework stated above, the simulation of the BPT for dioxin-like compounds is first performed by multiplying the LADD and the cancer slope factor (CSF). The CSF is defined to be the carcinogenic risk per unit of dose (a carcinogenic or potentially carcinogenic substance). That is, the multiplication of the $LADD_{ijt}$ and $CSF_{dioxin}^{t}$ results in the risk of cancer $r_{ijt}$, listed in Equation (1) below:(1)$$r_{ijt}=LADD_{ijt}\times CSF_{dioxin}^{t}$$
where *i* indicates different age groups, *j* indicates the region, and *t* can be a specific point in time (year) or a time period (range of years). Equation (1) shows that the CSF is the same for people of different ages. The purpose of categorizing different age groups in the health risk assessment is to reflect the dissimilar LADDs through discriminated dietary patterns for each age group, which in turn will have a differentiated impact on health risk.

The expected number of people at risk for cancer $er_{ijt}$ at time *t* is then the product of the risk of cancer $r_{ijt}$ for different age groups living within different regions of the country and the exposed population $pop_{ijt}$, as in Equation (2):(2)$$er_{ijt}=r_{ijt}\times pop_{ijt}$$

Therefore, the expected deaths from cancer at a specific time, $d_{ijt}$, is the product of the expected number of people at risk for cancer $er_{ijt}$ and the risk of mortality from cancer $m_{risk}^{t}$, as in Equation (3):(3)$$d_{ijt}=er_{ijt}\times m_{risk}^{t}$$

### 2.3. Data for Dioxin Settling Flux and Dioxin Emissions for Simulation in Each Region

To obtain the dioxin settling flux in this study, the diffusion and spread of dioxins estimated by the Taiwan EPA via the Weather Research and Forecasting (WRF)/Chem model were used. WRF/Chem was used to simulate the settling of dioxin in each county or city in 1999, 2007, and 2010 [53]. Daily emissions larger than 0.1 g-TEQ (toxic equivalent quangtity, hereafter TEQ) were selected as the input items in that simulation. According to the report, 79 counties and cities had emissions greater than 0.1 g-TEQ in 1999, but the number decreased to 27 counties and cities in 2010. Of the regions of Taiwan, the eastern region had only a few counties and cities with emissions greater than 0.1 g-TEQ; thus, annual dioxin settling was simulated for only those counties and cities. On the other hand, most counties and cities located in the central and southern regions had emissions higher than the above mentioned level. However, counties and cities in all four regions except for three islands outside the main island of Taiwan were included in the simulation.

The outcomes of that simulation essentially provide the results for the “Computation of total dioxin exposure in a country/region” from the left-hand side of Figure 1. The estimates of the annual settling of dioxin in that report are shown in Table 1, and these results will be used for additional purposes. Another report from the Taiwan EPA [54] further indicates that the most complete record of the total annual emissions of dioxins shows 57.8 g I-TEQ was emitted across the whole country in 2010. However, the exact amount of dioxin emissions in each region was not recorded by the Taiwan EPA. If governmental efforts are made to decrease the annual dioxin settling amount in each region, then the proportion of regional dioxin emissions relative to national emissions would reasonably be the same as the proportion of regional annual settling relative to national annual settling. The corresponding total amount of annual dioxin emissions that could be eliminated in each region are listed in Table 1.

### 2.4. Data for the LADD and CSF and the Simulated Risk of Cancer in Each Region

The above data on the settling of dioxin were further used by the Taiwan EPA to simulate the concentration of dioxin in all kinds of environmental media, such as food, water, and soil, through a multiple media transport model developed by the US EPA [53]. The total exposure effect from dioxin calculated by the Taiwan EPA is mainly based on exposure via ingestion and inhalation but does not account for the exposure effect from dermal absorption due to a lack of data.

The report also uses data from the Council of Agriculture on the average quantity of various kinds of food consumed by the Taiwanese population [55] and the inhalation data are from the Ministry of Health and Welfare, Taiwan [56]. The summation of the exposure effects from each of these pathways is the average LADD of dioxin-like compounds for each person in each different age category and each city and county listed, as shown in Table 2. In addition to the LADD associated with absorption, the CSF of 1.0 × 10^−4^ (pg-TEQ/kg/day)^−1^, suggested by the US EPA in 1994 [57] and used in the study of the Taiwan EPA, is used here in Equation (1) to compute the risk of cancer arising from the diffusion and spread of dioxins for the northern, central, southern, and eastern regions of Taiwan, as listed in Table 3.

The CSF as estimated by the US EPA is based on 2, 3, 7, 8-tetrachlorodibenzo-p-dioxin (TCDD), the most toxic dioxin-like substance in most experiments and/or assessments for the detection of the toxicity of dioxins. The TCDD is employed in the report accomplished by the Taiwan EPA. Since this study uses results from the report of the Taiwan EPA, TCDD is also the dioxin-like compound used in this study for further simulation.

### 2.5. Data on the Projected Total Population for 2021–2070 in Each Region

According to the newest available dioxin reduction effort accomplished by the Taiwan EPA for 10 years from 2011 to 2020, it is known that the average annual reduction rate of dioxin-like compounds was 1.603% [58]. If the reduction path follows past experience for 2021–2050, it will decrease by 48.09%. That is, the Taiwanese government has set a goal to reduce dioxin emissions by approximately 50% of their current level in each region by 2050. A much more ambitious emissions reduction goal is set to reduce them by 40% for the remaining 50% beyond 2050, i.e., for 2051–2070.

Simulating the expected number of people at risk for cancer due to dioxin emissions requires knowing the total population in each region. Total population projections for 2021–2070 were obtained from the National Development Council [59]. The size of the population is projected for the whole country by age but not by region. It is reasonable to assume that in the next 50 years, the share of the population in each region will remain the same as it has been over the past 20 years, 2001–2020. The reason for using the population proportion for each region is that the total projected population for Taiwan by the end of 2021 is less than that in 2020. This phenomenon is predicted to continue until 2070. That is, the total size of the population is projected to decrease by approximately 25% by the end of 2070. Thus, it is inappropriate to use the population growth rates that held in the past for the population projections for the years 2021–2070. The population proportion for each region is therefore used for this purpose.

Administrative divisions were restructured multiple times between 2001 and 2020. Period 1 covers 2001–2009, when some counties and cities were combined. Period 2 covers 2010–2013, when a county was merged with a municipality. In the most recent period, 2014–2020, four cities and counties were promoted to municipalities. The population proportions for the northern, central, southern, and eastern regions in period 2 and period 3 were almost the same. However, to account for the long-time trend of regional population proportions, 20-year averages of these proportions are used. The northern region accounts for 44.48% of the population, the central region accounts for 24.88%, the southern region accounts for 27.70%, and the eastern region has an average of 2.46% of the population [60]. With these population shares, the population of each region is projected for the three age-group classifications for 2021–2070 in 5-year intervals. These projections are listed in Table 4.

### 2.6. Simulation of the Expected Number of People at Risk for Cancer and Expected Cancer Mortality

Once the population projections presented in Table 4 and the simulated cancer risk for each age group in each region in 2020 and each 5-year interval from 2021 to 2070, presented in Table 3, are calculated, the expected number of people at risk for cancer in each age group, region, and 5-year interval is determined.

On the basis of the simulated expected number of people at risk for cancer, a high risk of death from cancer is expected. The coefficient for the risk of cancer mortality is required to simulate the expected cancer mortality. The coefficient for the risk of cancer mortality is collected from the Taiwan Cancer Registry [61] which reports the 5-year cancer survival rates for Taiwan. As dioxin-like compounds remain in the environment for decades or even centuries, their impacts on food chains and hazards to human and animal health are cumulative. It is normally hard to differentiate one type of cancer from the other unless a specific case causal relationship is examined. The average cancer survival rate across all types of cancer can properly reflect the benefits of emission reductions in dioxin-like compounds in a broad scope such as this study intends to achieve. From these data, the one-year cancer survival rate for all types of cancer from 2013 to 2017 was 77.22%. The corresponding cancer mortality risk was 22.78%. By using Equation (3), the expected cancer mortality rate is then estimated.

### 2.7. Monetization of the Cancer Mortality Risk by Conversion to Current Values

The use of the value transfer method (VTM) is necessary to adopt the VSL calculations from Liou’s study [62]. The study done by Liou [62] was based on the hedonic wage method (HWM), a revealed preference method. The HWM, using readily available wage data from the labor market and resulting in a low operating cost, is the most dominant approach in the estimation of VSL. The result from Liou [62] is the newest VSL estimation in Taiwan. We then employed Equation (5) in this study to transfer the VSL value from 2014 to 2020 and for further time periods henceforth. Value conversions are essential in order to standardize monthly earnings and price levels across different years. The earnings used to calculate the 2014 VSL adopted from Liou’s study [62] are adjusted as shown in Equation (4):(4)VSL2020=VSL2014×dijt1+ϵW×(W2020−W2014W2014)100
where $VSL_{2020}$ is the adjusted nominal VSL for 2020, $\epsilon_{W}$ is the earnings elasticity estimated by Liou [62], and $W_{2020}$ and $W_{2014}$ are the average monthly earnings for 2020 and 2014. All the variables are defined in Table 5.

The next step is to monetize the cancer mortality risk by using the data collected above. The BPT for dioxin is computed under the assumption of a uniform distribution of dioxin emissions across regions. It is essential to use Equation (4) to monetize expected cancer mortality. In Equation (4), the most important variable is VSL. Liou [62] is referenced for the computation of the VSL in this study. Only a few studies have estimated the VSL for Taiwan. As the VSL changes significantly over time and as Liou’s study [62] is the most recent study available, it best reflects current conditions.

The price level adjustment is shown in Equation (5):(5)$$RVSL_{2020}=VSL_{2014}\times (CPI_{2014}/CPI_{2020})$$
where $RVSL_{2020}$ is the real VSL, which uses the consumer price index (CPI) to account for inflation. $CPI_{2020}$ and $CPI_{2014}$ are the consumer price indices for 2020 and 2014. The estimated real VSL that accounts for changes in wages is US$2,194,866 for children under 14 years old, US$12,326,614 for adults 15–64 years old (the majority of the labor force), and US$25,745,461 for people of retirement age, 65 years old or above. We use the average exchange rate over the past 20 years, i.e., 2001–2020, to convert the New Taiwan dollar to the US dollar at the average US$ to TWD exchange rate from over the past 20 years, 2001–2020, of 1:31.807.

With the above information, the BPTs for the different regions, different age groups, and Taiwan as a whole are ready to be estimated for the 2021–2070 projected populations. In this estimation, the use of the 20-year average exchange rate, i.e., from 2001 to 2020, is much more appropriate than the use of a single-year average. The VSL for each age group from 2021 to 2070 is then estimated. The VSLs for each of the projected years are discounted by 2%, the 20-year real rate of return on government bonds minus the inflation rate proposed by the Taiwan EPA [39]. The results are listed in the top portion of Table 6. The VSL for 2020 and all the discounted VSLs listed in Table 6 for each age group over every 5-year interval is the average VSL over each age included in the group and each year during the 5-year interval. For example, the VSL for children in 2021–2025, i.e., US$2,167,689 per person, is the summation of the VSL for newborns to those aged 14 in 2021–2025 divided by 75. A similar procedure was followed for all other age groups and 5-year intervals.

## 3. Results

### 3.1. Monetization of Expected Cancer Mortality

Once all variables and parameters are prepared and VSLs have been calculated, the results of the monetary value of expected cancer mortality $mdc_{ijt}$, shown in the bottom portion of Table 6, can be calculated by multiplying the VSLs from the top portion of Table 6 and the corresponding expected cancer mortality for 2020 and the 5-year intervals from 2021 to 2070 with a mortality risk of 22.78%. The potential loss of life is calculated with Equation (6), and the results are listed in the bottom portion of Table 6.
(6)$$mdc_{ijt}=d_{ijt}\times VSL_{ijt}$$

The magnitude of $mdc_{ijt}$ indicates the potential monetary value of the loss of life for persons in each different age group living in each different region for every 5-year interval from 2021 to 2070.

### 3.2. Computation of the BPT from the Elimination of Dioxin Emissions by Age Group, Region, and Time Period

The BPT from a reduction in dioxin emissions for each age group in a given region and time period, $BPT_{ijt}$, is calculated by dividing $mdc_{ijt}$ by the amount of dioxin emissions in the region and time period $E_{jt}$, as in Equation (7):(7)$$BPT_{ijt}=mdc_{ijt}/E_{jt}$$
where $BPT_{ijt}$ indicates the benefit for a specific age group in a given region and time period when there are $E_{jt}$ dioxin emissions in time period *t*. However, it is normally meaningless to compute the BPT for only one particular age group, as dioxin emissions do not target only those of a specific age. Thus, the BPT for each different region in each specific time period $BPT_{ijt}$ is reasonably computed as shown in Equation (8):(8)$$BPT_{jt}=\overset{n}{\underset{i=1}{\sum }}mdc_{ijt}/E_{jt}$$
where $\sum_{i=1}^{n}mdc_{ijt}$ is the monetary value of expected cancer mortality for a specific age group. This is calculated by summing the values of $mdc_{ijt}$ for each particular age group in each region in a specific time period for the case when dioxin emissions in that particular region and time period have been eliminated.

Depending upon the complexity of the analytical or policy requirements, BPT can also be computed without differentiating across ages and regions. That is, $BPT_{t}$ is a weighted average calculated by summing over *n* age groups and *k* regions for a given time period for the case when the corresponding dioxin emissions have been reduced, as shown in Equation (9):(9)$$BPT_{t}=\sum_{i=1}^{n}\sum_{j=1}^{k}mdc_{ijt}/\sum_{j=1}^{k}E_{jt}$$

### 3.3. Net BPT When Dioxin Emissions Have Been Reduced

As total dioxin emissions in Taiwan are currently measured in grams, the BPT calculated in Equation (8), $BPT_{jt}$, is the monetary value of the BPT for the case when dioxin emissions in a particular region and specific time period are reduced. The monetary value in Equation (9), $BPT_{t}$, indicates that a specific amount of dioxin emissions has been eliminated from a particular region. All these results are computed by taking the values from the bottom portion of Table 6 and dividing them by the values in the top portion of Table 7. If actions to reduce emissions are not taken, then future dioxin emissions in each region will remain at their 2020 level, as shown in the bottom portion of Table 7. The BPT for the case when emissions are not reduced is calculated by dividing the values in the bottom portion of Table 6 by the values in the bottom portion of Table 7.

## 4. Discussions

### 4.1. Policy Implications from the Results

The total net BPTs in each region and Taiwan as a whole for every 5-year interval from 2021 to 2070 are presented in Table 8 and Figure 2. The results clearly show that the total net BPT for the southern region of Taiwan is higher than that for Taiwan as a whole. In each 5-year period, the total net BPT for the northern region is the second highest, the central region is third, and the eastern region is the lowest. Providing decision makers with information on these BPTs is essential for them to set appropriate standards for reducing dioxin emissions in each region of Taiwan. These results clearly indicate that the net BPTs from a reduction in dioxin-like emissions in Taiwan as a whole for every 5-year period beyond 2050, i.e., 2051–2070, are 5.34 times those for each 5-year period in 2021–2050.

The current policies implemented by the Taiwan EPA for managing dioxin-like compounds are basically command-and-control for setting the same standard for the entire country. That is, one standard fits all. As the monetary benefits evaluated by this study show, for every 5-year specific time period, the total net BPT for the southern region is the highest, the northern region is the second highest, the central region is the third highest, and the eastern region is the lowest. The results indicate that, without reducing dioxin-like compounds, the damage will be highest for the southern region and lowest for the eastern region. The efficient command-and-control policy implies that it is appropriate to set different standards for different regions, assuming that the cost of reduction is the same for all regions. Thus, the results from this study imply that the standard should be most stringent for the southern region and more relaxed for the eastern region.

### 4.2. BPT Comparison among Regions and across Age Groups

The total net BPT for the three age groups from a reduction in emissions of dioxin-like compounds increases progressively over each 5-year period from 2051 to 2070 for both Taiwan as a whole and each region. This finding is evident from the net BPT curves shown in Figure 2, with those for 2051–2070 being much steeper than those for 2021–2050 in all regions. These results imply that a lack of action now and continuing into the future will be regretted. If all reduction efforts are postponed until close to a half-century from now, more expenditures to prevent the large loss of health from cancer due to the spread and diffusion of dioxins and dioxin-like compounds will be inevitable.

The details of the BPTs per gram from a reduction in emissions of dioxin-like compounds in each region for each age group and 5-year period from 2021 to 2070 are shown in Figure 3. Comparison of net BPT for a given age group among different regions in a 5-year period in either 2021–2050 or 2051–2070 is thus possible. Specifically, the total net BPTs for the first 50% reduction in emissions of dioxin-like compounds in 2021–2050 generate gains of US$2,697,448 per gram for the country as a whole. The total net BPT of a 40% reduction in the remaining emissions creates gains of US$14,087,483. These results show that additional reductions in emissions of dioxin-like compounds can create more than five times the benefit of the initial reductions.

This outcome demonstrates that without further efforts to reduce emissions of dioxin-like compounds, there will be a very large loss of health due to cancer mortality. Similarly, the total net BPTs from a reduction in emissions of dioxin-like compounds for the northern, central, southern, and eastern regions of Taiwan for each 5-year period from 2021 to 2050 are 6.31, 4.74, 5.31, and 4.66 times those for each 5-year period in 2021–2050, respectively. The total net BPTs for the first 50% reduction in emissions of dioxin-like compounds in 2021–2050 generate gains of US$1,369,356 per gram, US$891,195 per gram, US$3,972,444 per gram, and US$96,599 per gram for the northern, central, southern, and eastern regions of Taiwan, respectively. The total net BPT of a 40% reduction in the remaining emissions creates gains of US$8,647,030 per gram, US$4,225,620 per gram, US$21,077,961 per gram, and US$449,694 per gram for the northern, central, southern, and eastern regions, of Taiwan, respectively.

### 4.3. Uncertainty Analyses of Key Parameters

The framework was constructed in this study to demonstrate how the benefits from reductions in dioxin-like compounds are evaluated. The novelty of this framework is the combining various steps of data (parameters) of physical processes from previous studies with socio-demographic data, such as VSL and population. It is a complicated process to evaluate the impact of dioxin-like compound emissions. As ambitions further evaluate the benefits of dioxin-like compounds emission reduction, the change of essential parameters in the entire process could result in different benefit per ton outcomes. The results presented in Table 6 are evaluations of all related parameters from determined and designated outcomes of the existing studies. However, once the essential parameter is different from that used in Table 6 for BPT, the result for every unit of dioxin-like compounds emission reduction might be different.

The percentage change in the impact of BPT by way of this uncertainty is conducted below for each key parameter. These results of BPT compared with BPT of US$ 837,915 per gram, measured by a percentage higher or lower than those in Table 6, are presented in Table 9. Once the BPT of dioxin-like compound reductions is evaluated for each key parameter under different magnitudes, all the corresponding net benefits in each region for each age group for each 5-year period will be changed accordingly. Different results deemed as the measurement of uncertainty or different results from the selection of different parameters is debatable. However, the provision of all possible BPTs influenced by all the key parameters is able to break the image that BPT is fixed in a stationary status. The information below is for the possible BPT values compared to that shown in Table 6 for a change in each key parameter. Such outcomes are broadly viewed as uncertain results if it is too bold to treat them as uncertainty.

First of all, the LADD is simulated by the Taiwan EPA [5]. A Monte Carlo type of analysis is included in that study to account for uncertainty. The final estimated result presented is for the 95th percentile of the LADD distribution. However, no other factor of the transfer function was found in that report. Thus, it was unable to carry out further Monte Carlo analysis without detailed input parameters for further simulation. That report stated that the reason for taking the 95th percentile is to account for uncertainty in a relatively conservative way. That is, it would rather treat the impact of a LADD seriously than belittle its influence. Secondly, the magnitude of CSF from the US EPA [57] is from integrating various studies of cohorts and obtaining a consensus from experts’ judgments. The final magnitude is suggested in a conservative manner. No other information is available that is similar to the problem for further simulation than that provided by the LADD.

As for the other two key parameters related to sociodemographic attributes, one is VSL and the other is population. If VSL uses the estimation value of a 90% or 95% confidence interval, other than the mean value of Liou’s study [62], the BPT will then have a ±11.5% and ±13.5% difference from the use of the VSL mean. The final key parameter is the population of different age groups. There are high, low, and medium-variant projected populations for the next 50 years since the estimation of BPT in 2021. These projected results are generated by [59]. The projected population in Table 6 is the medium-variant projection. Assuming that the low-variant or the high-variant projected population is adopted, respectively, for the first and the last 5-year period, i.e., 2021–2025 and 2066–2070, the BPT evaluation for the children group in 2021–2025 is different from the medium-variant projection by −0.065% to 0.042%. When the BPT estimation is conducted for a distant time period, 2066–2070, the result for this group with a low-variant or high-variant projected population is different from the medium-variant population projection by −2.847% to 3.193%. When a similar low-variant and high-variant projected population are used, the BPT results for the working-age adults group is not different from the medium-variant projected population adopted in 2021–2025 but has a −0.0004% to 0.0005% difference in the distant future, 2066–2070. As with the older adults, the estimated BPT will not have a difference when the low or high-variant projected population is adopted; although the estimated BPT is different with the low or high-variant projected population, it can almost be disregarded or neglected.

The uncertainty test of the population parameter indicates that the late marriage and low fertility rate in Taiwan have an immediate impact on the children group population. Thus, the low, medium, or high-variant population projection is mainly reflected in the children group, as Taiwan has been an aging society for years. Different variants of the projected population will not make any difference for the group of old adults. This, in turn, will have an influence on the benefit estimation of the BPT of dioxin-like compound reductions. The impact of utilizing different variants of the projected population on working-age adults on the estimation of BPT will be deferred and not be observed in immediate years, such as 2021–2025. Moreover, it can be concluded that different magnitudes of VSL have the most impact on BPT estimation among four key parameters. The number of working-age adults and children in the immediate or in the distant future are the other two parameters that generate notable uncertainty.

## 5. Conclusions

This study follows the framework established by the US EPA for calculating the BPT from a reduction in dioxin emissions in Taiwan. The results indicate a BPT of US$837,915 per gram of dioxin each year. This result can be explained in two ways: It can be interpreted as the cost to society of every unit (gram) of dioxin emitted, i.e., the cost imposed on society, or it can be interpreted as the benefit to society for each one-gram reduction in dioxin emissions, i.e., a benefit to society.

The BPT is calculated as the monetary value of the benefit from a reduction in emissions of dioxin-like compounds from 2021 to 2050 for children aged 0–14, working-age adults aged 15–64, and older adults aged 65 or above living in the northern, central, southern, and eastern regions of Taiwan on the basis of population projections for 2021–2070. The goal for the first 30 years, i.e., by 2050, is for dioxin-like compound emissions to be reduced 50% of their current level. Currently, population projections for Taiwan are available until 2070. The BPT between 2051 and 2070 is calculated for the case in which the remaining 50% of the emissions are reduced by 40% by 2070. If actions to reduce emissions of dioxin-like compounds to these target levels are undertaken, then the total net BPT for the whole country is projected to be US$78,282 per gram for the 2021–2025 period. The total net BPT for the southern, northern, central, and eastern regions is projected to be US$111,716 per gram, US$62,506 per gram, US$62,506 per gram, and US$3,464 per gram, respectively, for the 2021–2025 period. Assuming that reductions in the emission of dioxin-like compounds are implemented in the current period, the projected total net BPT per gram is clearly the highest for the southern region and higher for this region than for Taiwan as a whole. This result is consistent for each 5-year period from 2026 to 2070.

Determining the benefits from reductions in emissions of dioxin-like compounds, as computed and simulated in this study, i.e., the BPTs per gram, for different age groups of residents in each region and in different time periods is essential to identify emission-reduction priorities. The budget for implementing any public program is always limited. If optimization of the net benefit is a major concern, the rational approach to selecting a project is to choose the one that can potentially generate the highest possible benefit. This is a criterion for efficiently allocating all kinds of limited resources by the government. This study advances the measurement of the reduction in emissions of dioxin-like compounds from a physical assessment to a monetary benefit when a cost–benefit analysis is enforced. Furthermore, conducting a benefit-cost analysis for a command-and-control type policy in managing dioxin-like compounds constitutes yet another potential area of study. Such a critical topic deserves further research efforts. The evaluation framework for dioxin-like compounds and the results proposed and provided in this study are useful, containing essential information regarding monetary benefits which is rarely accomplished in the literature but is accounted for in the benefit-cost analysis of a command-and-control policy for dioxin-like compounds.

Certain limitations exist regarding the methodologic calculation of BPT for dioxin. The calculation of BPTs for emission reductions in dioxin-like compounds is generated under an assumption of uniformity, i.e., there is no regional differentiation in the reduction in emissions of dioxin-like compounds; the results cannot reflect any regional discrepancies. This means that whenever one unit of dioxin is emitted in Taiwan, the BPT per gram is the same regardless of where it is reduced. The other limitation is related to some coefficients and parameters used in this study. The coefficients of the diffusion and spread of dioxin-like compounds are from 2010. Moreover, the CSF derived from human cohort data with a magnitude 1.0 × 10^−4^ was the most current accessible data obtained from the US EPA in 1994; however, the assessment was conducted by the Taiwan EPA in 2013. This study uses simulation outcomes from the Taiwan EPA for further evaluation which is an inevitable limitation of using the method of international parameter transfer. Notwithstanding the existence of these limitations, BPT calculations based on updated coefficients and/or parameters are recommended in order to clarify existing conditions. Moreover, the test of the uncertainty of four key parameters finds that the determination of the magnitude of a statistical life, the number of children in the immediate time period and the foreseeable future, and the number of working-age adults in a distant time period will have an attentive impact on the estimation of BPT. This indicates that understanding the potential key factors with a noticeable impact on the estimated outcome is essential, especially in the case of a complicated process. Additionally, the delicate treatment of uncertainty is another dimension worthy of further exploration.

## Figures and Tables

**Figure 1 ijerph-19-06701-f001:**
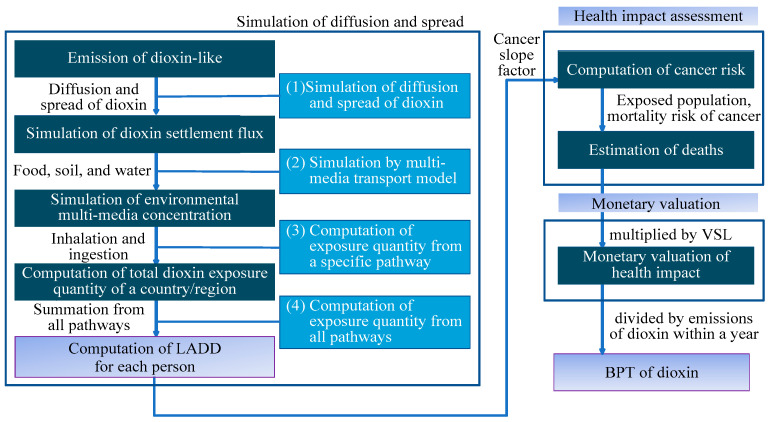
The impact pathway approach framework for assessing the monetary BPT of dioxin emissions.

**Figure 2 ijerph-19-06701-f002:**
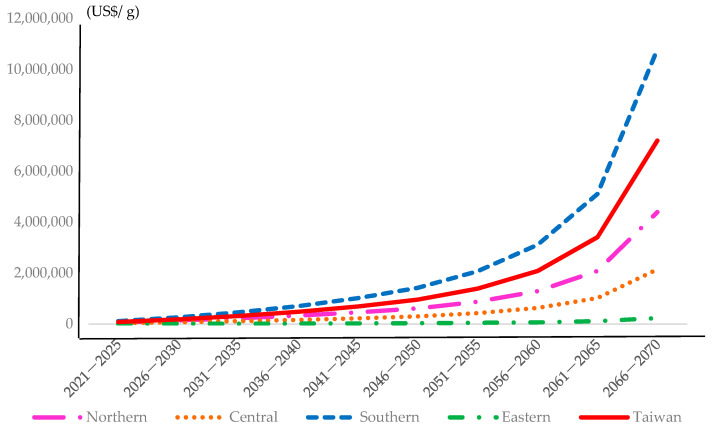
Net BPT for each region and the whole county under the implementation of dioxin-emission reduction action.

**Figure 3 ijerph-19-06701-f003:**
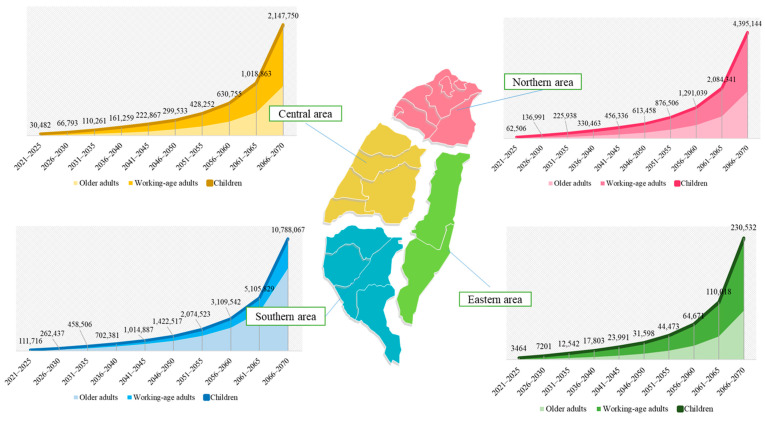
Total net BPT over 5-year periods in 2021–2050 and 2051–2070 for specific age groups in different regions.

**Table 1 ijerph-19-06701-t001:** Annual dioxin settling simulated by the Taiwan EPA for each region and the corresponding distribution of dioxin emissions across regions. Source: [53].

Settling or Emissions	Region			
	Northern	Central	Southern	Eastern
Annual dioxin settling (pg I-TEQ/m^2^/year)	12.40	41.70	68.80	1.47
Annual dioxin emissions (g I-TEQ/year)	5.7628	19.3798	31.9743	0.6832

Note: The Taiwan EPA did not simulate dioxin settling for three small islands (counties)—Kinmen county, Lianjing county, and Penghu county—that are located outside the main island of Taiwan since the formation and sources of dioxin settling for these islands differ from those on the main island. Thus, these three islands are excluded from all subsequent analyses.

**Table 2 ijerph-19-06701-t002:** LADD data for different age groups in different regions. Source: [53].

Age Group (Age)	Region				Taiwan
	Northern	Central	Southern	Eastern	
Children (0–14)	7.26 × 10^−4^	2.24 × 10^−3^	9.21 × 10^−3^	8.37 × 10^−5^	3.19 × 10^−5^
Working-age adults (15–64)	1.52 × 10^−3^	4.44 × 10^−3^	1.56 × 10^−2^	1.82 × 10^−4^	6.00 × 10^−3^
Older adults (65 or above)	7.68 × 10^−4^	2.27 × 10^−3^	2.79 × 10^−2^	7.79 × 10^−5^	8.84 × 10^−3^

**Table 3 ijerph-19-06701-t003:** The simulated risk of cancer for each age group in each region. Source: [53].

Age Group (Age)	Region				Taiwan
	Northern	Central	Southern	Eastern	
Children (0–14)	7.26 × 10^−8^	2.24 × 10^−7^	9.21 × 10^−7^	8.37 × 10^−9^	3.19 × 10^−7^
Working-age adults (15–64)	1.52 × 10^−7^	4.44 × 10^−7^	1.56 × 10^−6^	1.82 × 10^−8^	6.00 × 10^−7^
Older adults (65 or above)	7.68 × 10^−8^	2.27 × 10^−7^	2.79 × 10^−6^	7.79 × 10^−9^	8.84 × 10^−7^

**Table 4 ijerph-19-06701-t004:** Actual total population in 2020 and projected total population in each region over 5-year intervals from 2021 to 2070. Source: [59].

Region/ Age Group	Year										
	2020	2021 –2025	2026 –2030	2031 –2035	2036 –2040	2041 –2045	2046 –2050	2051 –2055	2056 –2060	2061 –2065	2066 –2070
**Northern**											
Children	1,318,119	1,264,892	1,192,701	1,124,171	1,132,252	1,113,024	1,064,712	1,001,184	943,498	899,735	871,704
Working-age adults	7,477,322	7,267,601	6,877,924	6,504,056	6,081,763	5,624,320	5,156,703	4,821,029	4,488,235	4,156,714	3,911,141
Older adults	1,684,598	1,922,894	2,326,148	2,665,299	2,908,197	3,120,420	3,290,247	3,287,280	3,246,798	3,183,865	3,021,274
Total	10,480,038	10,455,388	10,396,773	10,293,526	10,122,212	9,857,764	9,511,662	9,109,492	8,678,531	8,240,315	7,804,120
**Central**											
Children	737,293	707,520	667,140	628,808	633,328	622,573	595,549	560,015	527,748	503,269	487,590
Working-age adults	4,182,459	4,065,151	3,847,184	3,638,060	3,401,850	3,145,977	2,884,415	2,696,655	2,510,505	2,325,069	2,187,707
Older adults	942,284	1,075,576	1,301,137	1,490,842	1,626,707	1,745,415	1,840,408	1,838,748	1,816,105	1,780,903	1,689,957
Total	5,862,036	5,848,247	5,815,461	5,757,710	5,661,885	5,513,965	5,320,372	5,095,417	4,854,358	4,609,241	4,365,254
**Southern**											
Children	820,861	787,714	742,757	700,080	705,112	693,138	663,051	623,489	587,565	560,312	542,855
Working-age adults	4,656,515	4,525,912	4,283,239	4,050,412	3,787,429	3,502,555	3,211,346	3,002,304	2,795,056	2,588,601	2,435,670
Older adults	1,049,086	1,197,486	1,448,613	1,659,820	1,811,085	1,943,247	2,049,007	2,047,159	2,021,949	1,982,758	1,881,504
Total	6,526,462	6,511,111	6,474,609	6,410,312	6,303,626	6,138,940	5,923,405	5,672,953	5,404,571	5,131,671	4,860,030
**Eastern**											
Children	72,900	69,956	65,963	62,173	62,620	61,557	58,885	55,371	52,181	49,761	48,210
Working-age adults	413,539	401,940	380,389	359,712	336,357	311,057	285,195	266,631	248,225	229,890	216,309
Older adults	93,168	106,347	128,649	147,406	160,840	172,577	181,970	181,805	179,567	176,086	167,094
Total	579,606	578,243	575,001	569,291	559,817	545,191	526,050	503,807	479,973	455,737	431,613
**Taiwan**											
Children	2,963,396	2,843,731	2,681,433	2,527,363	2,545,531	2,502,302	2,393,687	2,250,863	2,121,174	2,022,787	1,959,767
Working-age adults	16,810,525	16,339,032	15,462,958	14,622,428	13,673,029	12,644,604	11,593,307	10,838,644	10,090,456	9,345,131	8,793,034
Older adults	3,787,315	4,323,054	5,229,648	5,992,129	6,538,212	7,015,332	7,397,137	7,390,467	7,299,455	7,157,971	6,792,433
Total	23,561,236	23,505,817	23,374,039	23,141,920	22,756,772	22,162,238	21,384,132	20,479,973	19,511,085	18,525,889	17,545,234

Note: Neither the total population in 2020 nor the projected populations for 2021–2070 include those counties off the main island of Taiwan, i.e., Kinmen, Lianjing, and Penghu counties.

**Table 5 ijerph-19-06701-t005:** Relevant parameters for the VSL conversion.

Parameter	Value	Definition
$VSL_{2014}\;{}^{a}$	11.79	The VSL computed in Liou’s study in 2019 [62]; unit: millions of US$
$\epsilon_{W}\;{}^{b}$	0.2476	The earnings elasticity estimated for monthly earnings between US$1317.18 and US$1580.61
$W_{2014}\;{}^{c}$	1317.18	Average monthly earnings in US$ in 2014
$W_{2020}\;{}^{c}$	1385.64	Average monthly earnings in US$ in 2020
$CPI_{2014}$	98.93	$\mathrm{The}\;\mathrm{consumer}\;\mathrm{price}\;\mathrm{indices}{\;}^{d}\;\mathrm{for}\;2014\;\mathrm{and}\;2020,\;\mathrm{with}\;\mathrm{the}\;\mathrm{base}\;\mathrm{year}\;\mathrm{of}\;2015;\;i.e.,\;CPI_{2015}=100$
$CPI_{2020}$	102.55	

Note ^a^: This study uses the study by Liou [62] as a reference. The US$–TWD exchange rate was 1:30.368 in 2014. ^b^: The VSLs from Liou’s study [62] are used to compute the elasticity results used in this study. ^c^: Data on *W_t_* are from the Directorate-General of Budget, Accounting, and Statistics, Executive Yuan, R.O.C., Taiwan [63]. ^d^: *CPI_t_* values are from the Directorate-General of Budget, Accounting, and Statistics, Executive Yuan, R.O.C., Taiwan [64].

**Table 6 ijerph-19-06701-t006:** Estimated VSLs for the different population age groups in 2020 and the projected population age groups every 5 years from 2021 to 2070.

Age Group/ Region	Year										
	2020	2021 –2025	2026 –2030	2031 –2035	2036 –2040	2041 –2045	2046 –2050	2051 –2055	2056 –2060	2061 –2065	2066 –2070
**VSL (US$/person)**											
Children	2,194,866	2,167,689	2,151,162	2,134,532	2,117,808	2,100,999	2,084,114	2,067,906	2,050,987	2,034,013	2,016,989
Working-age adults	12,326,614	12,173,987	12,081,168	11,987,772	11,893,849	11,799,449	11,704,618	11,613,595	11,518,578	11,423,246	11,327,640
Older adults	25,745,461	25,426,682	25,232,819	25,037,751	24,841,583	24,644,418	24,446,355	24,256,242	24,057,790	23,858,678	23,658,995
**Estimated total monetary value of death from cancer due to dioxin emissions for each age group for 2020 and each 5-year interval from 2021 to 2070 (US$)**											
**Northern**											
Children	47,848	45,305	,378	39,702	39,603	38,658	36,680	34,327	31,995	30,307	29,045
Working-age adults	3,191,360	3,062,975	2,877,734	2,699,646	2,504,845	2,297,353	2,090,445	1,938,309	1,789,987	1,643,805	1,533,762
Older adults	759,491	854,337	1,026,976	1,166,759	1,264,437	1,345,585	1,408,110	1,394,734	1,366,482	1,328,928	1,251,561
**Central**											
Children	82,527	78,254	73,139	68,518	68,405	66,812	63,357	59,142	55,172	52,274	50,223
Working-age adults	5,214,158	5,005,944	4,700,782	4,411,500	4,092,673	3,754,585	3,414,237	3,167,027	2,924,567	2,686,747	2,506,807
Older adults	1,253,804	1,413,724	1,698,169	1,930,411	2,089,177	2,225,391	2,327,293	2,306,769	2,259,026	2,197,384	2,067,796
**Southern**											
Children	377,956	358,319	335,151	313,563	313,224	305,485	289,900	270,482	252,887	239,200	229,735
Working-age adults	20,398,082	19,580,641	18,388,746	17,255,199	16,007,931	14,686,774	13,357,311	12,390,544	11,441,404	10,508,244	9,805,205
Older adults	17,167,073	19,352,248	23,231,857	26,412,324	28,595,147	30,438,321	31,836,488	31,559,797	30,916,666	30,066,706	28,291,426
**Eastern**											
Children	219	217	215	213	212	210	208	207	205	203	202
Working-age adults	20,955	20,696	19,330	17,982	16,651	15,339	14,046	12,775	11,519	11,423	10,195
Older adults	5149	5085	5047	7511	7452	7393	7334	7277	7217	7158	7098
**Taiwan**											
Children	472,555	447,845	419,261	392,113	391,794	381,962	362,427	338,309	316,057	299,000	287,219
Working-age adults	28,322,862	27,186,948	25,533,549	23,958,761	22,227,225	20,392,987	18,547,138	17,204,379	15,886,423	14,590,912	13,613,558
Older adults	19,636,063	22,136,469	26,572,682	30,213,054	32,706,429	34,815,170	36,415,290	36,100,566	35,362,545	34,389,899	32,360,774

**Table 7 ijerph-19-06701-t007:** The reduction in dioxin emissions for each 5-year interval from 2021 to 2070.

Region	Year										
	2020	2021 –2025	2026 –2030	2031 –2035	2036 –2040	2041 –2045	2046 –2050	2051 –2055	2056 –2060	2061 –2065	2066 –2070
The amount of dioxin emissions with a reduction plan for each region (g)											
Northern	5.7628	5.2826	4.8023	4.3221	3.8419	3.3616	2.8814	2.3051	1.7288	1.1526	0.5763
Central	19.3798	17.7648	16.1498	14.5348	12.9198	11.3049	9.6899	7.7519	5.8139	3.8760	1.9380
Southern	31.9743	29.3097	26.6452	23.9807	21.3162	18.6517	15.9871	12.7897	9.5923	6.3949	3.1974
Eastern	0.6832	0.6262	0.5693	0.5124	0.4554	0.3985	0.3416	0.2733	0.2050	0.1366	0.0683
Taiwan	57.8000	52.9833	48.1667	43.3500	38.5333	33.7167	28.9000	23.1200	17.3400	11.5600	5.7800
The amount of dioxin emissions without a reduction plan for each region (g)											
Northern	5.7628	5.7628	5.7628	5.7628	5.7628	5.7628	5.7628	5.7628	5.7628	5.7628	5.7628
Central	19.3798	19.3798	19.3798	19.3798	19.3798	19.3798	19.3798	19.3798	19.3798	19.3798	19.3798
Southern	31.9743	31.9743	31.9743	31.9743	31.9743	31.9743	31.9743	31.9743	31.9743	31.9743	31.9743
Eastern	0.6832	0.6832	0.6832	0.6832	0.6832	0.6832	0.6832	0.6832	0.6832	0.6832	0.6832
Taiwan	57.8000	57.8000	57.8000	57.8000	57.8000	57.8000	57.8000	57.8000	57.8000	57.8000	57.8000

**Table 8 ijerph-19-06701-t008:** Net BPT values of dioxin emissions reduction for different age groups in each region for 2020 and projected values for every 5-year period in 2021–2070 *.

Age Group/Region	Year										
	2020	2021 –2025	2026 –2030	2031 –2035	2036 –2040	2041 –2045	2046 –2050	2051 –2055	2056 –2060	2061 –2065	2066 –2070
**Northern**											
Children	0	715	1471	2296	3436	4792	6365	8935	12,955	21,035	45,358
Working-age adults	0	48,315	99,877	156,154	217,323	284,758	362,748	504,530	724,782	1,140,927	2,395,247
Older adults	0	13,476	35,643	67,488	109,704	166,786	244,345	363,041	553,301	922,379	1,954,538
Northern Total	0	62,506	136,991	225,938	330,463	456,336	613,458	876,506	1,291,039	2,084,341	4,395,144
**Central**											
Children	0	367	755	1179	1765	2462	3269	4578	6643	10,789	23,323
Working-age adults	0	23,483	48,513	75,879	105,593	138,383	176,175	245,130	352,122	554,539	1,164,150
Older adults	0	6632	17,525	33,204	53,902	82,021	120,089	178,545	271,990	453,535	960,276
Central Total	0	30,482	66,793	110,261	161,259	222,867	299,533	428,252	630,755	1,018,863	2,147,750
**Southern**											
Children	0	1019	2096	3269	4898	6824	9067	12,689	18,454	29,924	64,666
Working-age adults	0	55,673	115,023	179,887	250,325	328,092	417,754	581,275	834,938	1,314,576	2,759,959
Older adults	0	55,024	145,317	275,351	447,158	679,971	995,696	1,480,558	2,256,149	3,761,330	7,963,442
Southern Total	0	111,716	262,437	458,506	702,381	1,014,887	1,422,517	2,074,523	3,109,542	5,105,829	10,788,067
**Eastern**											
Children	0	29	63	104	155	220	305	454	700	1191	2658
Working-age adults	0	2757	5661	8773	12,192	16,040	20,558	28,045	39,328	66,905	134,344
Older adults	0	678	1,478	3,665	5,456	7,731	10,735	15,975	24,643	41,922	93,531
Eastern Total	0	3464	7201	12,542	17,803	23,991	31,598	44,473	64,671	110,018	230,532
**Taiwan**											
Children	0	704	1451	2261	3389	4720	6270	8780	12,759	20,692	44,723
Working-age adults	0	42,761	88,351	138,170	192,278	252,014	320,885	446,480	641,320	1,009,752	2,119,758
Older adults	0	34,817	91,947	174,239	282,928	430,241	630,022	936,866	1,427,554	2,379,924	5,038,875
Taiwan Total	0	78,282	181,748	314,671	478,595	686,975	957,177	1,392,126	2,081,633	3,410,368	7,203,356

Note *: The unit for all the magnitudes in this table is US$ per gram of dioxin emissions reduced.

**Table 9 ijerph-19-06701-t009:** The impact on BPT from each key parameter.

Key Parameter	Impact on BPT Estimation ^a^	Source of Uncertainty and/or Different Selections of Key Parameter
LADD ^b^	----	-----
CSF ^c^	----	-----
VSL	±11.5% to ±13.5%	The uncertainty is from the selection of VSL by selecting a 90% confidence interval and 95% confidence interval instead of the mean value used in Table 6.
**Population**		
Children	2021–2025: −0.065% to 0.042% 2066–2070: −2.847% to 3.193%	The uncertainty for the group of children is from the use of a low-variant projected population and high-variant projected population of the 5-year period in 2021–2025 and in 2066–2070 instead of the medium-variant projected population used in Table 6.
Working-age adults	2021–2025: 0% 2066–2070: −7.576% to 6.404%	The uncertainty for the group of working-age adults is from the use of low-variant projected population and high-variant projected population of the 5-year period for 2021–2025 and 2066–2070 instead of the medium-variant projected population used in Table 6.
Older adults	2021–2025: 0% 2066–2070: −0.0004% to 0.0005%	The uncertainty for the group of older adults is from the use of a low-variant projected population and a high-variant projected population of the 5-year period for 2021–2025 and 2066–2070 instead of the medium-variant projected population used in Table 6.

Note ^a^: The impact is shown in percentage change of BPT higher or lower than US$837,915 per gram of dioxin emission reductions each year obtained in the main text with deterministic magnitudes. ^b^: Further Monte Carlo simulation cannot be carried out due to a lack of input parameters in the original transfer function. The detailed reason is stated in the main text. ^c^: The magnitude of CSF from the US EPA [57] is from integrating various studies of cohorts, and obtaining a consensus from experts’ judgments. The final magnitude is suggested in a conservative manner. No other information is available for further simulation.

## Data Availability

Data is not publicly available. The data can be obtained on the request of the authors.

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
