# Peer review of "The Monetary Benefits of Reducing Emissions of Dioxin-like Compounds—Century Poisons—Over Half a Century: Evaluation of the Benefit per Ton Method"

_ijerph, 2022, doi:10.3390/ijerph19116701_

Round 1

Reviewer 1 Report

Environmental chemical pollutants have an important impact on public health. The impact on people of different ages are obvious differences. The monetary cost and monetary benefit of pollution emission reduction can more intuitively measure the practical benefits of emission reduction actions and policy implementation. This research evaluated the monetary value of the health benefits on the dioxin-like compounds emissions reduction in the next 50 years. It also carried out comparative analysis for different age structures, different regions and different development periods. The research methods and conclusions are rigorous and scientific. The research results have practical value and application significance. This theme is an important scientific issue in the field of environmental science and public health research. The revised manuscript has reached the IJERPH standard and could be accepted for publish.

This manuscript is a resubmission of an earlier submission. The following is a list of the peer review reports and author responses from that submission.

Round 1

Reviewer 1 Report

Major comments

  • In order to be of sufficient interest for journal publication, a study of the benefits of reducing dioxin-like compounds needs a more innovative methodology and a more interesting policy application. Is there a specific policy being considered in Taiwan for dioxin reductions, and could you estimate the benefits and the costs of this policy? Could you consider a Monte Carlo analysis to evaluate the benefits and costs of the policy accounting for uncertainty about the key parameters used in the analysis? Are there countervailing health risks to society from the technologies or needed to reduce dioxin emissions?
  • Considering the critical importance of the cancer slope factor in the analysis, insufficient details are provided about this parameter and the underlying study from which it is derived (“CSF of 1.0x10-4 (pg- 279 TEQ/kg/day)-1 suggested by the US EPA in 1994”). Neither the original study estimating the cancer slope factor nor the 1994 US EPA report used to identify the cancer slope factor are even included in the list of references.
    • Which specific dioxins and dioxin-like compounds were included in the original study deriving the cancer slope factor? If it was not derived based on all dioxin-like compounds, why is it appropriate to apply the factor to all of these compounds?
    • Which age groups were included in the original study deriving the cancer slope factor? Is it appropriate to apply this cancer slope factor to all three of the age groups? This would seem unlikely, but if it is appropriate, why bother to separate out the three age groups?
    • The EPA report mentioned was published almost 30 years ago. Is the cancer slope factor still applicable to the current population? How to dioxin emissions evaluated in the original study compare to the Taiwanese population that the cancer slope factor is applied to?
  • Why use the average cancer survival rate across all types of cancer? What are the specific types of cancer shown to have a causal relationship or at least association with dioxin exposure?
  • More discussion is needed of the 2019 Liou et al. VSL study. VSL is another critical parameter used in this study. What method was used to estimate willingness to pay to avoid mortality risks? Stated preference or revealed preference? What years where the data collected?

Minor comments

  • 2 Unclear: “It is clear that the most important actions for the relevant agency or institute in any 50 country are to assess the changes in the levels of dioxin-like POPs under various activities 51 or circumstances and to examine whether dioxin emissions are above the limit set by the 52 nation or international agencies.”
  • 2 – The manuscript makes multiple references to “heavy metals” and seems to conflate studies on dioxins and heavy metals. Please clarify if the studies on heavy metals are relevant for understanding and quantifying impacts of dioxins, and if not I recommend deleting these.
  • 2-3 – don’t understand this statement: “The carcinogenic or noncarcinogenic human health risk is often used as an indicator of socioeconomic status among people who live in areas with different distributions of 95 PCDD/PCDF emissions or concentrations”
  • 3: what is the source for the statement that “dioxin-like compound emissions are expected to be reduced to half of their current level by 2050”? What is the connection between dioxide reductions and GHG targets for the year 2050?
  • 4 discusses the “optimal” level of environmental protection from dioxins. However, a socially optimal level of pollution release must be based on an accounting of marginal benefits and costs. The authors have not presented sufficient information to support the claim that the dioxin reductions assumed in their study are socially optimal.
  • 4 : Shouldn’t the first step of the impact path approach be to quantify changes in human health exposure per change in emissions? Only than can you quantify the change in human health impacts, and the finally monetize those impacts.
  • P 4 – figure 1 – “LADD” – please spell out/define
  • The definition of “cancer slope factor” appears to be incorrect - “cancer slope factor t CSFdioxin, a quantitative risk assessment parameter that measures the lifetime exposure of a specific agent.”

Author Response

Attached file is the reply to reviewer 1

Reviewer 2 Report

General comments

The paper is suitable for the Journal.

It is well organized, easy to read and understand.

The conclusions are based on research's findings.

Limits of the study are included.

Specific comments

It is obvious that the paper's findings are quite valuable for health and environmental policies in Taiwan.

But it could be useful to highlight more its value for the existing knowledge regarding the relationship between public health and pollution. This study advances the measurement of reductions of emissions of dioxin-like compounds  from physical assessment to monetary benefit when a cost–benefit analysis is enforced. This idea (of measurement) must be analyzed in the context of the existing literature.

Recommendations

A little bit more discussion is needed regarding how this study enriches the literature regarding public health protection and pollution.

More and concrete policy suggestions are required.

Author Response

Attached file is the reply to reviewer 2

Reviewer 3 Report

Environmental chemical pollutants have an important impact on public health. The impact on people of different ages are obvious differences. The monetary cost and monetary benefit of pollution emission reduction can more intuitively measure the practical benefits of emission reduction actions and policy implementation. This research evaluated the monetary value of the health benefits on the dioxin-like compounds emissions reduction in the next 50 years. It also carried out comparative analysis for different age structures, different regions and different development periods. The research methods and conclusions are rigorous and scientific. The research results have practical value and application significance. This theme is an important scientific issue in the field of environmental science and public health research. Some minor modification suggestions for improving the article.

  1. In the abstract, it is suggested to supplement the research background and significance, as well as modify the long sentence, so that the readers could understand the whole research more clearly.
  2. The result section contains many calculation processes, which can be integrated into the materials and methods
  3. According to monetary value of emissions reduction for different ages, regions and periods, it is suggested to discuss the targeted emission reduction actions and implementation policies, in order to provide practical reference for dioxin-like compounds emissions reduction.

Author Response

Attached file is the reply to reviewer 3

Round 2

Reviewer 1 Report

I appreciate the revisions made by the authors to address my specific concerns. In particular, the additional explanation about the cancer slope factor and the VSL estimate are very helpful. However, I remain concerned that the study lacks novelty of data and methods or a policy application of sufficient interest to readers. The authors' addition to the conclusion of discussion of topics for future research did not really address this overarching concern. 

Reviewer 2 Report

Accepted for publication.